



# Measurement report: Regional trends of stratospheric ozone evaluated using the MErged GRIdded Dataset of Ozone Profiles (MEGRIDOP)

Viktoria F. Sofieva[1], Monika Szelag[1], Johanna Tamminen[1], Erkki Kyrölä[1], Doug Degenstein[2], Chris Roth[2], Daniel Zawada[2], Alexei Rozanov[3], Carlo Arosio[3], John P. Burrows[3], Mark Weber[3], Alexandra Laeng[4], Gabriele Stiller[4], Thomas von Clarmann[4], Lucien Froidevaux[5], Nathaniel Livesey[5], Michel van Roozendael[6], Christian Retscher[7]

[1] Finnish Meteorological Institute, Helsinki, Finland

[2] Institute of Space and Atmospheric Studies, University of Saskatchewan, Saskatoon, Canada

[3] Institute of Environmental Physics, University of Bremen, Bremen, Germany

[4] Karlsruhe Institute of Technology, Institute of Meteorology and Climate Research, Karlsruhe, Germany

[5] Jet Propulsion Laboratory, California Institute of Technology, Pasadena, California, USA

[6] Royal Belgian Institute for Space Aeronomy (BIRA-IASB), Brussels, Belgium

[7] ESA/ESRIN, Frascati, Italy

*Correspondence to*: Viktoria F. Sofieva (viktoria.sofieva@fmi.fi)

## Abstract

In this paper, we present the MErged GRIdded Dataset of Ozone Profiles (MEGRIDOP) in the stratosphere with a resolved longitudinal structure, which is derived from data by six limb and occultation satellite instruments: GOMOS, SCIAMACHY and MIPAS on Envisat, OSIRIS on Odin, OMPS on Suomi-NPP, and MLS on Aura. The merged dataset was generated as a contribution to the European Space Agency Climate Change Initiative Ozone project (Ozone_cci). The period of this merged time series of ozone profiles is from late 2001 until the end of 2018.

The monthly mean gridded ozone profile dataset is provided in the altitude range from 10 to 50 km in bins of 10° latitude x 20° longitude. The merging is performed using deseasonalized anomalies. The created MEGRIDOP dataset can be used for analyses, which probe our understanding of stratospheric chemistry and dynamics. To illustrate some possible areas of applications, we created the climatology of ozone profiles with resolved longitudinal structure. We found zonal



asymmetry/structures in the climatological ozone profiles at middle and high latitudes associated with the polar vortex. At
Northern high latitudes, the amplitude of the seasonal cycle also has a longitudinal dependence.

The MEGRIDOP dataset has been also used to evaluate regional vertically-resolved ozone trends in the stratosphere,
including polar regions. It is found that stratospheric ozone trends exhibit longitudinal structures at Northern Hemisphere
middle and high latitudes, with enhanced trends over Scandinavia and Atlantic region. This agrees well with previous analyses
and might be due to changes in dynamic processed related to the Brewer-Dobson circulation.

## 1    Introduction


Nowadays, the importance of protecting the ozone layer and monitoring its recovery is well recognized (e.g.,
Petropavlovskikh et al., 2019; WMO, 2014, 2018).  Recent analyses indicated ozone recovery in the upper stratosphere (e.g.,
Arosio et al., 2019; Bourassa et al., 2014; Kyrölä et al., 2013; Petropavlovskikh et al., 2019; Sofieva et al., 2017; Steinbrecht
et al., 2017; WMO, 2018). The ozone recovery in the lower stratosphere is not yet observed, and the lower stratospheric ozone
trend is a subject of recent controversial discussions (Ball et al., 2018, 2019; Chipperfield et al., 2018).

In the majority of studies on ozone profile trends that use satellite observations made in limb-viewing geometry, analyses
are performed on zonal mean data. This representation allows ozone trends to be estimated globally. At the same time, such
representation provides a sufficiently large amount of experimental data in spatio-temporal bins (usually 10° latitude and one
month) to enable robust estimation of trends. This is especially important for the period before 2001, when long data records
are available only from solar occultation instruments having relatively scarce data coverage.

A recent study by Arosio et al. (2019) using the merged SCIAMACHY-OMPS dataset has shown that ozone trends for
the period 2003-2018 have a significant dependence on longitude. Also the trends of the total ozone column (WMO, 2018 and
references therein) have a pronounced zonal structure.

This paper is focused on a new longitudinally resolved merged dataset of ozone profiles in the stratosphere based on
several limb and occultation instruments. This new merged dataset is a contribution to the European Space Agency Climate
Change Initiative ozone project (Ozone_cci). It can be used in different applications, including the evaluation of regional ozone
trends in the stratosphere.

The paper is organized as follows. In Section 2, we briefly discuss the satellite data used for creating the merged dataset.
Section 3 is dedicated to the methodological aspects of data merging. Examples of ozone distributions are shown in Section 4.
Section 5 is dedicated to regional trend analysis. A discussion and summary (Section 6) conclude the paper.

## 2    Data

MEGRIDOP dataset is a merged and gridded dataset generated using the ozone profiles retrieved from several limb and
occultation instruments, viz. MIPAS (Michelson Interferometer for Passive Atmospheric Sounding), SCIAMACHY



(SCanning Imaging Spectrometer for Atmospheric CHartographY) and GOMOS (Global Ozone Monitoring by occultation of
Stars), all on Envisat, OSIRIS (Optical Spectrograph and InfraRed Imaging System) on Odin, OMPS-LP (Ozone Mapping and
Profiles Suite - Limb Profiler) on Suomi-NPP, and MLS (Microwave Limb Sounder) on Aura.

These instruments provide high-quality ozone profiles with a good vertical resolution of 2-4 km and a relatively dense
spatio-temporal coverage (100-3000 ozone profiles per day with a uniform sampling in longitude). The important information
about the datasets is collected in Table 1. More information about the datasets from the individual satellite instruments is found
in Petropavlovskikh et al., 2019, Sofieva et al., 2017 and references therein.

**Table 1. General information about the datasets.**

| Instrument/ satellite | Level 2 processor, references | Years | Vertical range/retrieval coordinate | Local time of Level 2 data | Number of profiles per day |
|---|---|---|---|---|---|
| MIPAS/Envisat | KIT/IAA V7R_O3_240 von Clarmann et al. (2003; 2009) | 2005-2012 | 6-70 km, Altitude | 10 a.m. and p.m. | ~1000 |
| SCIAMACHY/Envisat | UBr v3.5 (Jia et al., 2015) | 2002-2012 | 8-65 km, Altitude | 10 a.m. | ~1300 |
| GOMOS/Envisat | ALGOM2s v1 (Kyrölä et al., 2010; Sofieva et al., 2016) | 2002-2011 | 10-105 km, altitude | 10 p.m. | ~110 |
| OSIRIS/Odin | USask v5.10 (Bourassa et al., 2017; Degenstein et al., 2009) | 2001-present | 10-59 km, altitude | 6 a.m. and p.m | ~250 |
| OMLS-LP /SUOMI-NPP | USask 2D v 1.1.0 (Zawada et al., 2017) | 2012-present | 6- 59 km, altitude | 1:30 p.m | ~1600 |
| MLS/Aura | NASA v4.2 (Livesey et al., 2013) | 2004 - present | 261-0.02 hPa (~8-75 km), pressure | 1:30 a.m. and p.m. | ~3000 |

For all instruments except MLS, the original retrievals of ozone profiles are performed on an altitude grid. GOMOS,
OSIRIS, SCIAMACHY and OMPS - provide number density ozone profiles; therefore this representation (number density on
an altitude grid) is used for the merged dataset. For MIPAS, the retrievals are performed in volume mixing ratio vs. altitude
grid. The conversion to number density profiles is performed using temperature profiles retrieved by MIPAS and the pressure
profiles provided with the MIPAS ozone data, which are from the ERA-Interim reanalysis.

For MLS, retrievals are performed in mixing ratio on a pressure grid. Similarly to the conversion procedure of MIPAS
data, we performed the conversion to number density using the retrieved MLS temperatures, but for altitude-pressure
conversion, we used the ERA-Interim reanalysis data. Such conversion might introduce some uncertainty in the MLS data.
For studies of long-term changes, this uncertainty is associated with a potentially imperfect representation of temperature
trends in ERA-Interim, which might influence ozone trends. However, since current stratospheric temperature trends (after


2000) are small (Maycock et al., 2018; Steiner et al., 2020), this uncertainty is expected to be small. The MLS ozone profiles
data record is stable (Hubert et al., 2016), therefore including MLS data into the merged dataset is advantageous, especially
for the merging method applied in our work (see also below).

For all the instruments, we use the ozone profiles from the HARMonized dataset of Ozone profiles (HARMOZ_ALT)
developed in the ESA Ozone_cci project (Sofieva et al., 2013), https://climate.esa.int/en/projects/ozone/. HARMOZ consists
of the original retrieved ozone profiles from each instrument, which are screened for invalid data by the instrument experts
and are presented on a vertical grid (altitude-gridded profiles are used in our paper) and in a common netCDF4 format. The
detailed information about the original datasets can be found in (Sofieva et al., 2013), the references to the corresponding
publications are collected also in Table 1 of our paper.

## 3    Merging method

The method used for creating the MEGRIDOP dataset is similar to that used for the creation of the merged SAGE-CCI-OMPS
dataset (Sofieva et al., 2017). Below we describe and illustrate the merging process.

### 3.1    Gridded monthly mean from individual instruments

First, gridded ozone profile data $\rho_i(z,b,t)$ in 10°x20° latitude-longitude bins $b$ and at altitude $z$ were created for each
individual dataset $i$ and each month $t$. The mean number density profile in each spatio-temporal bin is termed $\rho_i(z,b,t)$. For
each instrument, we required more than 10 measurements in each spatio-temporal bin. The uncertainty of the averaged data
$\sigma_i(z,b,t)$ is characterized by the standard error of the mean. The non-uniformity of the sampling pattern can be characterized
by the inhomogeneity measure , which is defined as the linear combination of two classical inhomogeneity measures,
asymmetry $A$ and entropy $E$: $H = \frac{1}{2}(A+(1-E))$ (see Sofieva et al., 2014 for details). The unitless inhomogeneity measure $H$
ranges from 0 to 1 (the more homogeneous, the smaller $H$). For our application, we considered the main contribution to
sampling uncertainty, inhomogeneity in time $H_{time}$, only.

Examples of gridded dataset at 30 km altitude for individual satellite instruments are shown in Figures 1 and 2. All
instruments show a similar morphology, although biases between individual datasets exist. The coverage is instrument-specific
and to some extent time-dependent; the most complete coverage is achieved by MIPAS and MLS.





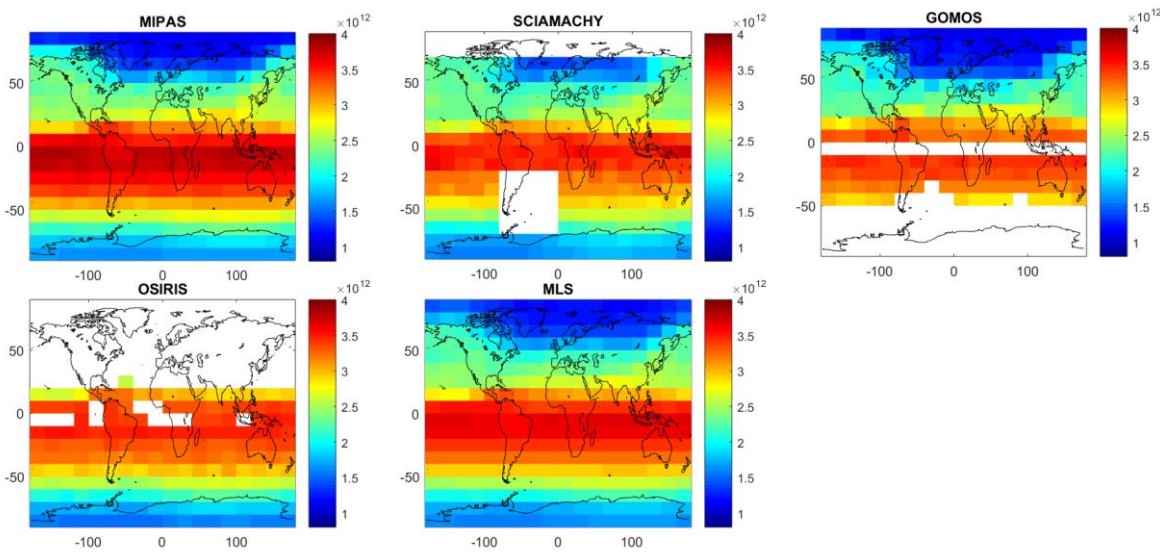

**Figure 1. Examples of gridded monthly mean ozonenumber density (cm⁻³) at 30 km for individual satellite instruments in January 2008**

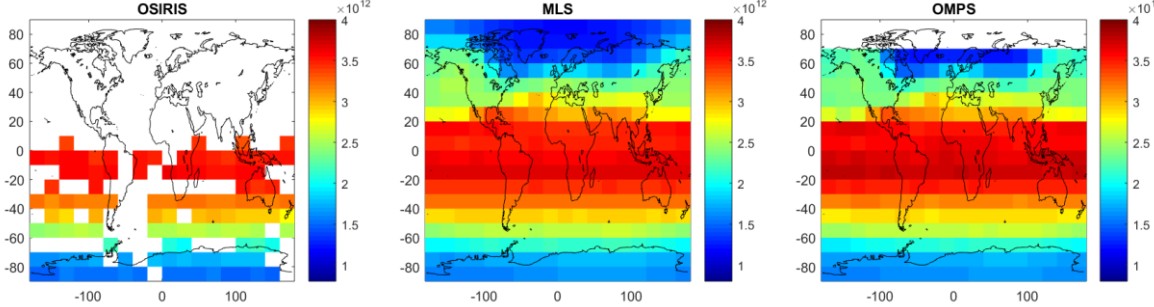

**Figure 2. Examples of gridded monthly mean ozone number density (cm⁻³) at 30 km for individual satellite instruments in January 2018.**

### 3.2    Seasonal cycle and deseasonalized anomalies

For each instrument $i$ , latitude-longitude bin $b$ , and altitude level $z$ , the deseasonalized anomalies are computed as:

$$\Delta_i(z,b,t) = \frac{\rho_i(z,b,t) - \rho_{m,i}(z,b)}{\rho_{m,i}(z,b)}, \qquad (1)$$





where $\rho_i(z,b,t)$ is the monthly mean value in this spatial bin and $\rho_{m,i}(z,b)$ is the climatological mean value for the month

$m$ . In other words, from each January we removed mean January values, from each February – the mean February value, and so on.

In our computations, we removed values for the spatial bins with less than 10 profiles and inhomogeneity larger than 0.9. For all instruments except for OMPS, the seasonal cycle is estimated using the years 2005-2011. For OMPS, the seasonal cycle is evaluated using the data from years 2012-2018. Figure 3 illustrates the seasonal cycle at 40 km for all instruments except

GOMOS, as the GOMOS data do not cover all months for the considered spatial bins. Although biases are visible, the overall behavior of the seasonal cycle is similar for the different datasets. In the tropics (left panel), small differences in seasonal cycle in two longitude regions, 0-20° E and 120-140°E are observed, while at mid-latitudes, all satellite instruments show consistently different seasonal cycles in the selected regions.

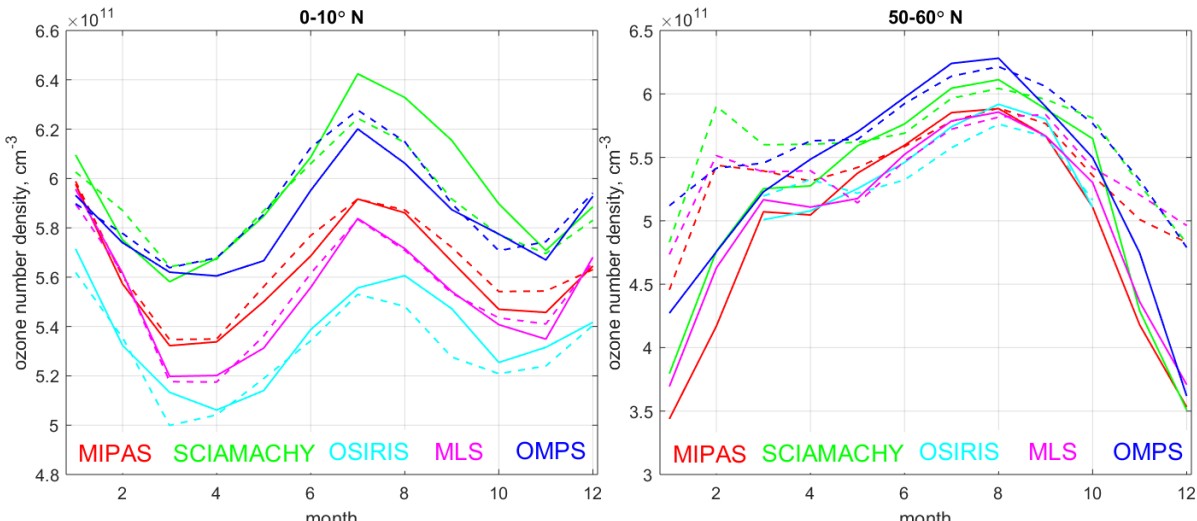

**Figure 3. Examples of seasonal cycles in the tropics (left) and NH upper stratosphere (right) at 40 km. Solid lines: longitudes 0-20°E, dashed lines: longitudes 120-140°E. In the tropics, a semi-annual cycle is observed.**

For two instruments - MIPAS and MLS - which measure during day and during night, and thus provide data at all latitudes in all seasons, we compared the relative amplitude of the seasonal cycle $\dfrac{\max(\rho_m) - \min(\rho_m)}{\text{mean}(\rho_m)}$ at several altitude levels (Figure




4). As seen from Figure 4, longitudinal structures in the relative amplitude of the seasonal cycle are observed, to be largest in the Northern middle and high latitudes, particularly in the middle and upper stratosphere.

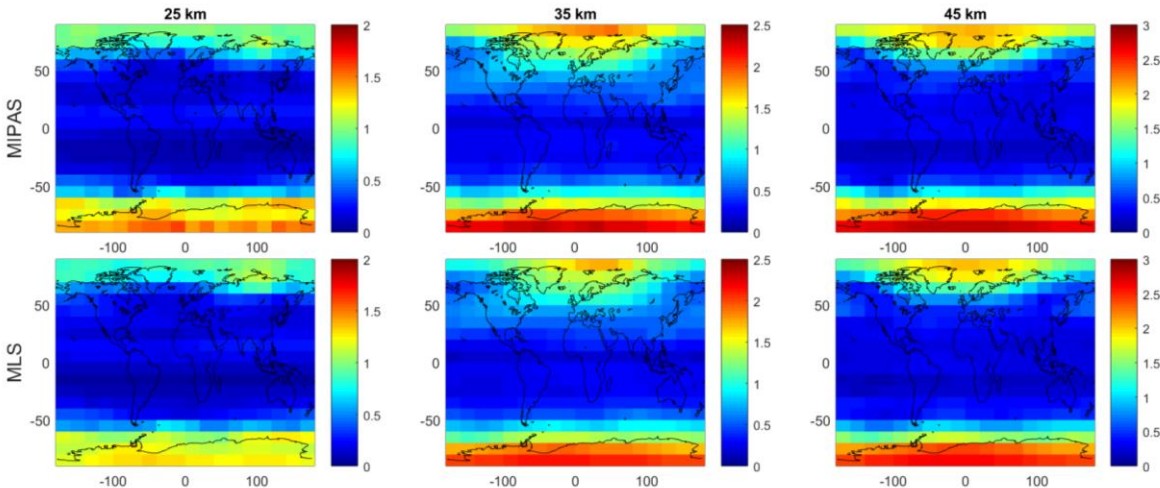

**Figure 4. Relative amplitude of seasonal cycle at 25 km (left), 35 km(center) and 45 km(right) for MIPAS (top) and MLS (bottom).**

The merging of individual datasets was performed on deseasonalized anomalies. The main advantage of using deseasonalized anomalies is various biases between individual datasets - instrumental, due to different sampling patterns, due to the difference in local time - are automatically removed, if the sampling patterns do not change over time. Details of the applied merging method are presented in the next section.

### 3.3     Merging the data

The merging method used for creating MERGRIDOP is similar to that used in creating the merged SAGE-CCI-OMPS dataset (Sofieva et al., 2017). The deseasonalized anomalies of all instruments except OMPS are aligned, as the seasonal cycle was estimated using the same period. First, we offset the OMPS deseasonalized anomalies to the median of the deseasonalized anomalies from all other instruments. These additive offsets are computed for the period 2012-2018, and the offsetting procedure is illustrated in Figure 5. In this figure, we selected a spatial bin where the effect of the offsetting is clearly visible.

In many other bins, the offsets are small or negligible. As observed in Figure 5 (and also below in Figure 6), the deseasonalized anomalies from individual datasets are in good agreement.



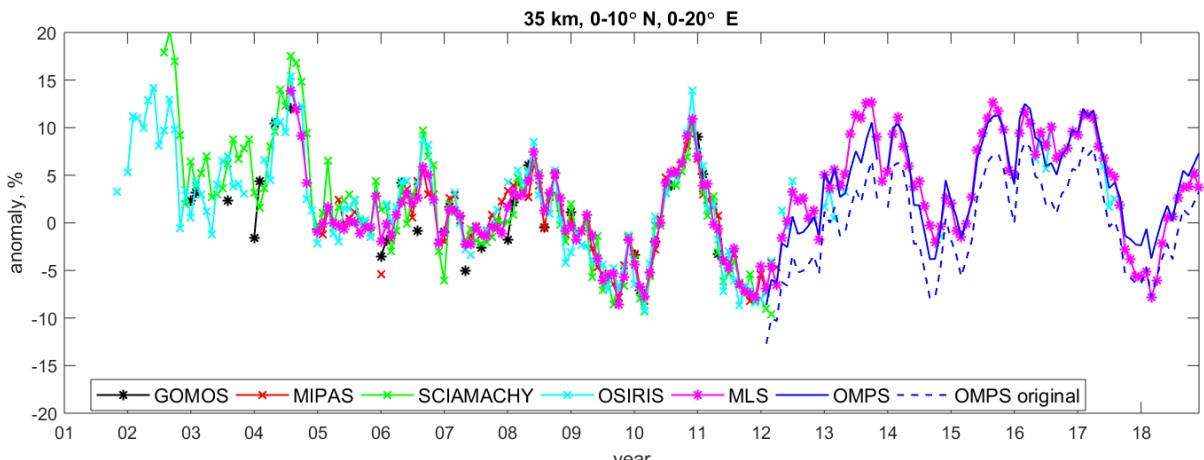

**Figure 5. Illustration of offsetting the OMPS deseasonalized anomalies. The data are shown for altitude 35 km and the**
**latitude/longitude bin 0-10° N/ 0-20° E.**

After offsetting OMPS, the merged ozone profiles in each spatiotemporal bin and at each altitude level is obtained from the

median of the deseasonalized anomalies corresponding to individual instruments:

$$\Delta_{merged}(z,b,t) = median(\Delta_i(z,b,t)) \tag{2}$$

The uncertainties of the merged deseasonalized anomalies are computed similarly to those used for the merged SAGE-CCI-

OMPS dataset (Sofieva et al., 2017). For each instrument, the uncertainty of the deseasonalized anomalies, $\sigma_{\Delta i}$, is given by

$$\sigma_{\Delta i} = \frac{1}{\rho_{m,i}}\sqrt{\sigma_i^2 + \sigma_{m,i}^2} \tag{3}$$

where $\sigma_i$ is the uncertainty of the gridded ozone profiles (see Sect. 3.1.) and $\sigma_{m.i}$ is the uncertainty of the seasonal cycle

$\rho_{m,i}$, which can be estimated via propagation of random uncertainties to the mean value:

$$\sigma_{m,i}^2 = \frac{1}{N_m^2}\sum_{j=1}^{N_m}\sigma_i^2(z,b,t_j) \tag{4}$$

Analogously to (Sofieva et al., 2017), the uncertainties of the merged deseasonalized anomalies are estimated as:

$$\sigma_{\Delta,merged} = min\left( \sigma_{\Delta,i_{med}}, \sqrt{\frac{1}{N}\sum_{i=1}^{N}\sigma_{\Delta,i}^2 + \frac{1}{N^2}\sum_{i=1}^{N}\left(\Delta_i - \Delta_{merged}\right)^2} \right), \tag{5}$$

where $\sigma_{\Delta,i_{med}}$ is the anomaly uncertainty of the instrument corresponding to the median value. Analogously to uncertainty

estimates in the merged SAGE-CCI-OMPS dataset (Sofieva et al., 2017), the uncertainties given by Eq. (5) can be interpreted





as follows. If individual anomalies are significantly different, the uncertainty of the merged anomaly is the uncertainty corresponding to the median value. In cases where several instruments report a similar anomaly (intersecting error bars), this provides more confidence in this anomaly value, and the resulting uncertainty of the merged anomaly is approximated by the second term in Eq. (5).

Examples of deseasonalized anomalies and their estimated uncertainties are displayed in Figures 6 and 7, respectively.


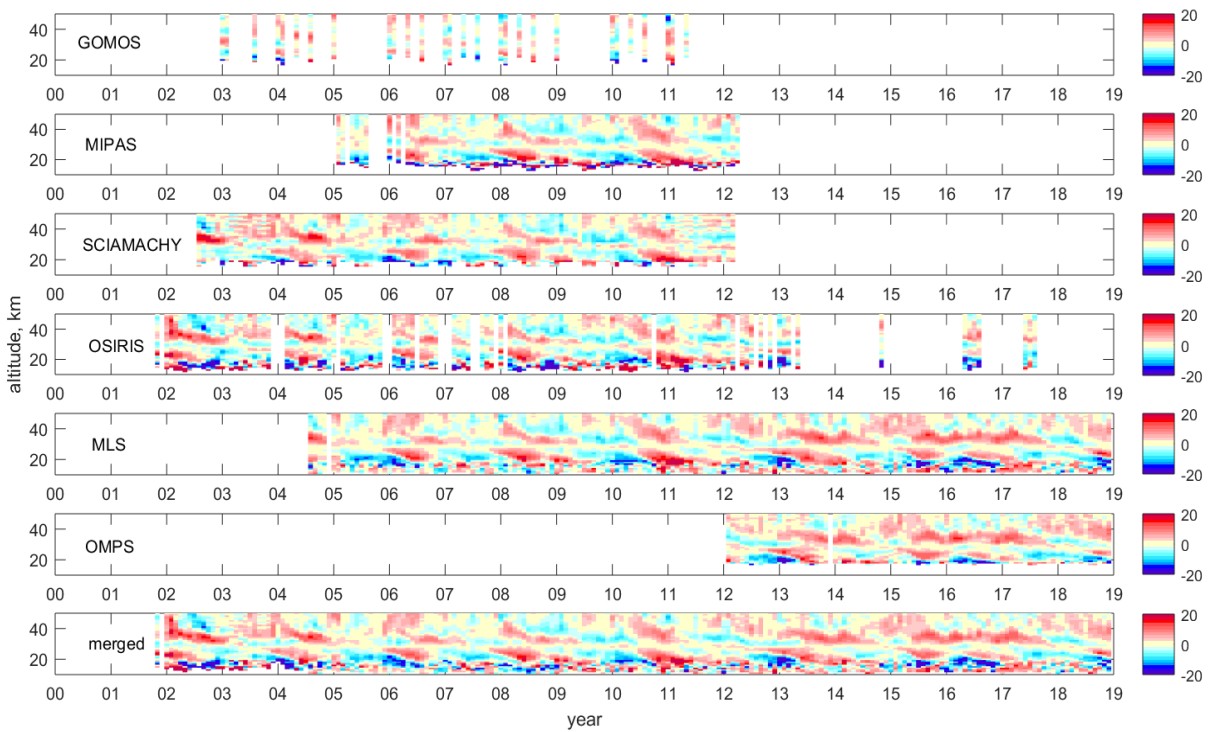

**Figure 6 . An example of deseasonalized anomalies (in %) for individual instruments and the merged dataset in the spatial bin 0-10° N, 0-20° E.**



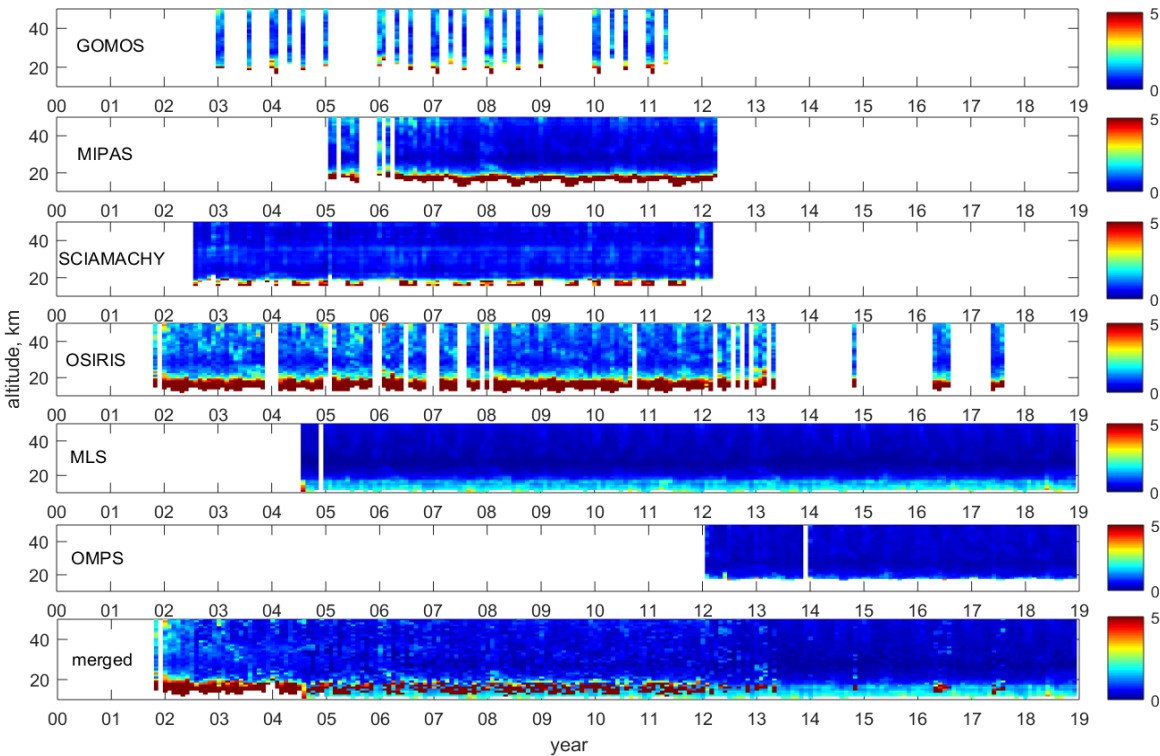

**Figure 7. An example of uncertainties in deseasonalized anomalies (in %) for individual instruments and the merged dataset in the spatial bin 0-10° N, 0-20° E**

## 4    The merged dataset and selected examples

The merged deseasonalized anomalies can be directly used for evaluation of ozone trends in the stratosphere. The

evaluation of regional ozone trends is discussed in Section 5 of our paper. We also created a version of MEGRIDOP in number

density through restoration of the seasonal cycle. This was achieved in a manner similar to that applied in creating the merged

SAGE-CCI-OMPS dataset (Sofieva et al., 2017). The best estimates of the amplitude and morphology of the seasonal cycle

are provided by MIPAS and MLS, as these two instruments provide global coverage in all seasons. The ozone profiles from

OSIRIS and MLS have the smallest biases with respect of ozone soundings (Hubert et al., 2016). For the seasonal cycle of the

merged dataset, we computed the mean of MIPAS and MLS seasonal cycles and offset it to the mean of OSIRIS and MLS

values (this offset does not depend on season). By this procedure, the seasonal cycle in the merged dataset has absolute values,

which   have the smallest biases with respect the ground-based instruments, and a realistic amplitude. An example of a number

density MERGRIDOP dataset is shown in Figure 8.





The merged dataset allows us to provide a gridded climatology of ozone profiles, i.e., the collection of ozone profiles

categorized by calendar month, latitude, longitude, and altitude. Figure 9 shows these climatological ozone values, for four

months and at four altitude levels. As observed in Figure 9, there are zonal asymmetry/structures associated with the polar

vortex, in both hemispheres. In other locations, the ozone distributions are rather uniform in longitude.

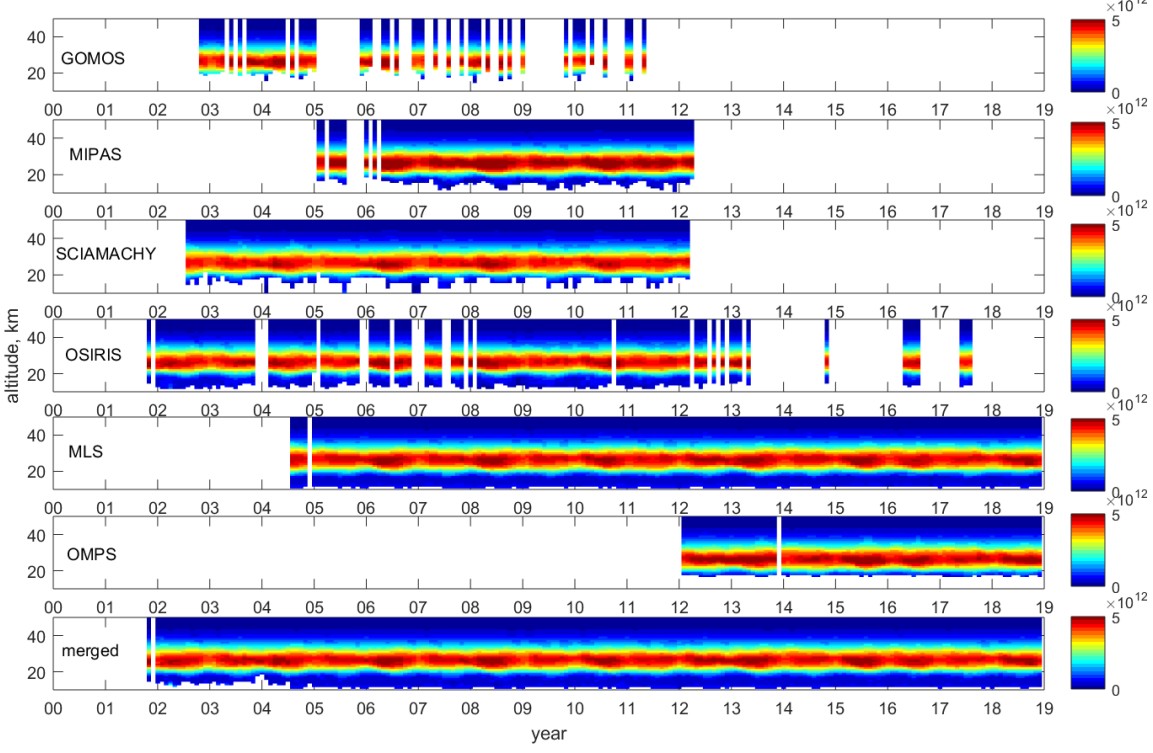

**Figure 8. An example of number density ozone profiles (in cm⁻³) for individual instruments and the merged dataset in the spatial bin**

**0-10° N, 0-20° E.**





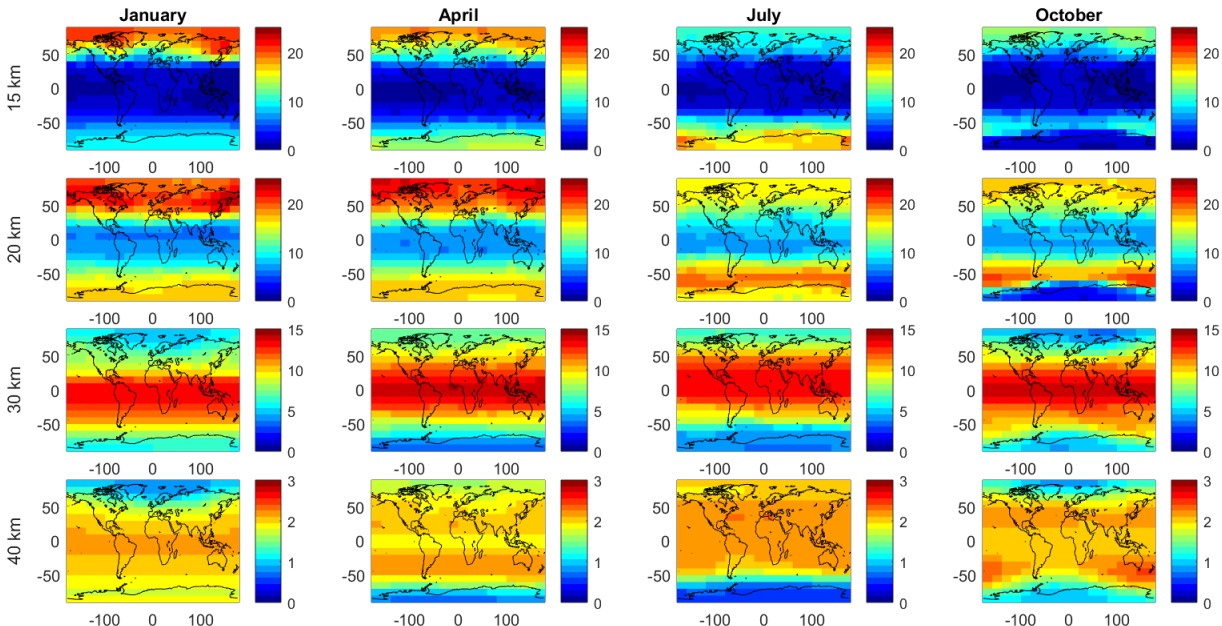

**Figure 9. Climatological ozone distributions (in DU/km), for January, April, July, and October, for selected altitude levels (15, 20, 30, and 40 km).**

## 5 Evaluation of regional ozone trends

For evaluation of the regional ozone trends, we exploited the standard approach of multiple linear regression and applied it to the deseasonalized anomalies:

$$\Delta_{merged}(t) = at + b + q_1 QBO_{30}(t) + q_2 QBO_{50}(t) + s\,F_{10.7}(t) + d\,ENSO(t)\,, \qquad (6),$$

where we model the trend with a simple linear term, $QBO_{30}(t)$ and $QBO_{50}(t)$ are the equatorial winds at 30 hPa and 50 hPa, respectively (http://www.cpc.ncep.noaa.gov/data/indices/), $F_{10.7}(t)$ is the monthly average solar 10.7 cm radio flux

(ftp://ftp.geolab.nrcan.gc.ca/data/solar_flux/monthly_averages/), and $ENSO(t)$ is the 2 month lagged ENSO proxy (http://www.esrl.noaa.gov/psd/enso/mei/table.html). The evaluation of trends has been performed for each latitude-longitude bin and for each altitude level separately. Autocorrelations are removed using the Cochrane–Orcutt transformation (Cochrane and Orcutt, 1949).

In our analysis, we consider long-term trends over the years covered by MEGRIDOP, and approximate them by a

linear function (which describes bulk changes). However, real changes in the atmosphere can be non-linear (Laine et al., 2014): if consider variations on a shorter scale, they can be different from long-term trends (e.g., (Arosio et al., 2019; Chipperfield et al., 2018; Galytska et al., 2019). We selected the years after 2003, in order to avoid the influence of major



sudden stratospheric warming in September 2002 on ozone trends at Southern Hemisphere middle and high latitudes (see also a discussion below).

Ozone trends (expressed in percent per decade) estimated at several altitude levels for years 2003-2018 are shown in Figure 10. Figure 11 displays the trends at these altitudes in absolute units, DU/(km decade). In Figures 10 and 11, black stars indicate the statistically significant trends, i.e., different from zero at a 95% or greater confidence level. The morphology of ozone trends presented in absolute and in relative units looks similar.  As shown in Figures 10 and 11, statistically significant trends are observed in the upper stratosphere. A longitudinal structure is clearly visible in the NH mid-latitude trends above

40 km: the trends are significantly larger over Scandinavia/Atlantic Ocean (5-6 % decade[-1]) than over Siberia (~1 % decade[-1]). The same feature was also observed by Arosio et al. (2019). Enhanced ozone trends over the mid-latitude Atlantic sector are seen in both absolute and relative units, and also at lower altitudes (but the ozone trends are not statistically significant below 40 km).

     We compared also the trends in late 2004 – 2018, the common measurement period, using MEGRIDOP, only MLS

data and the merged SCIAMACHY-OMPS dataset by Arosio et al. (2019). We found that the spatial distributions of ozone trends are similar for the considered datasets (Figure 12, top). The MEGRIDOP and pure MLS ozone trends in 2004-2018 are similar (as expected, MLS data are used in MEGRIDOP). SCIAMACHY-OMPS trends are somewhat larger, which might be related to the OMPS drift (Kramarova et al., 2018), but within error limits, and the morphology of ozone trends is similar. Specifically interesting is a two-core structure of ozone trends at NH polar region, which is seen nearly at all altitude levels

(Figure 12, bottom), for all datasets.

     There are several analyses showing that the residual circulation has a pronounced longitudinal two-core structure at Northern Hemisphere high and middle latitudes (e.g., Demirhan Bari et al., 2013; Kozubek et al., 2015). Kozubek et al. (2015) performed also the analysis of trends and have shown changes in the two-core structure of meridional winds. Arosio et al. (2019) suggested that this longitudinal structure in the NH mid-latitude ozone trends is due to changes in dynamical processes

related to the 3D structure of the Brewer Dobson circulation. However, the origin of the longitudinal structure of ozone trends requires a more detailed investigation, including simulations with chemistry-transport models, in future.

     Statistically significant (at 95% confidence level) positive trends (1-2 % decade[-1]) are observed also at SH mid-latitudes (~40°-50°S) at 25 km. This is in agreement with other studies of zonally averaged ozone trends (e.g., Arosio et al., 2019; Petropavlovskikh et al., 2019; Sofieva et al., 2017). In our analysis, there is a zonal asymmetry with larger trends in the

sector 50°W - 10°E.





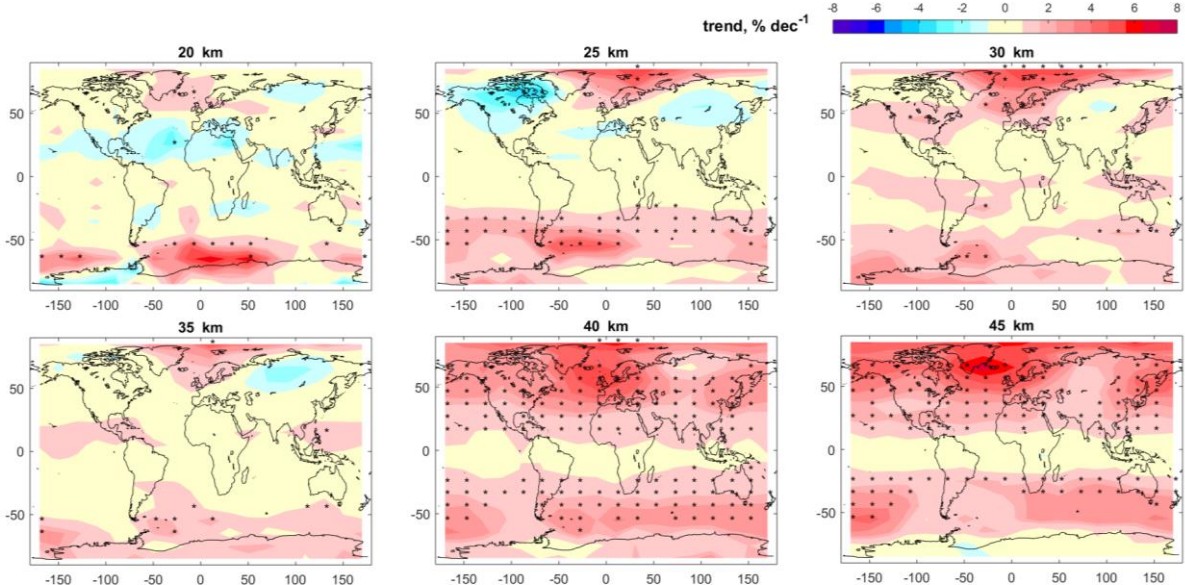

**Figure 10. Ozone trends (% decade$^{-1}$) in 2003-2018, for several altitudes. Statistically significant trends are indicated by stars.**

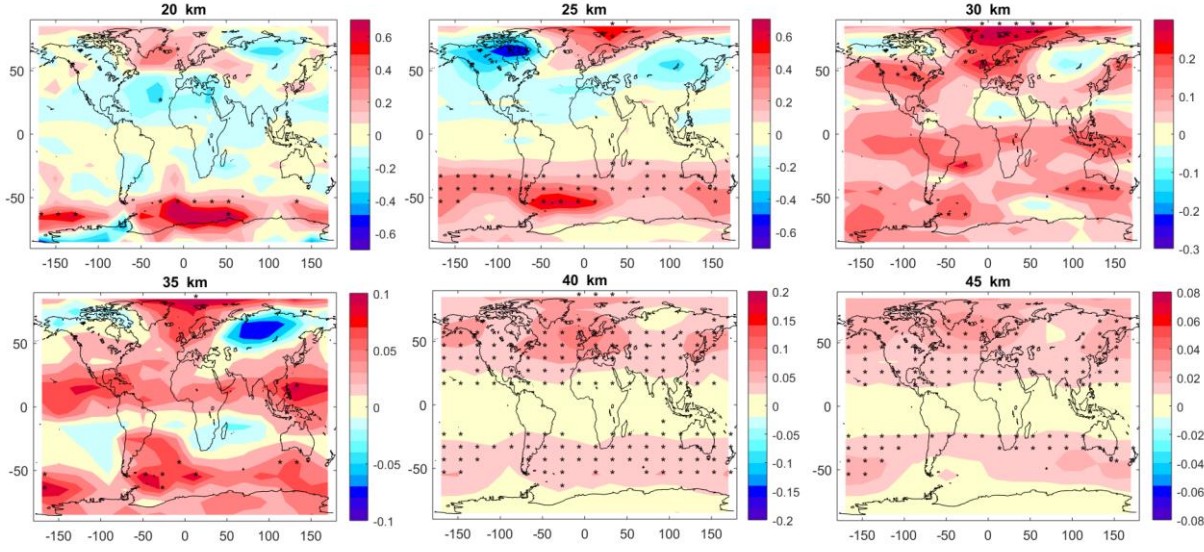

**Figure 11. Same as Figure 10, but trends in DU km$^{-1}$ decade$^{-1}$.**



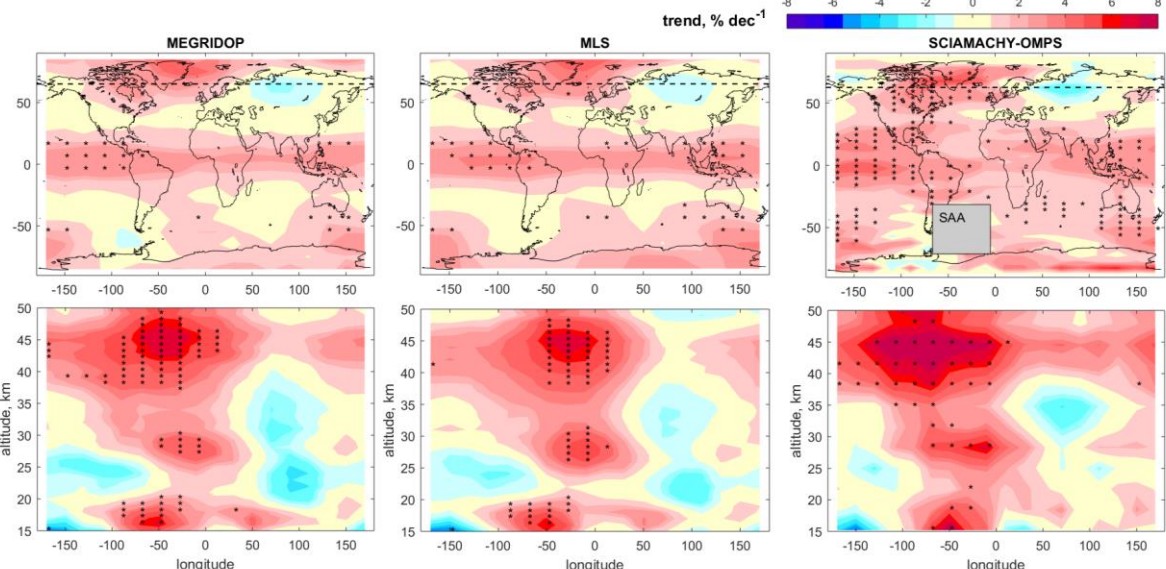

**Figure 12. Top: ozone trends in late 2004- 2018 (% decade-1) at 35 km, bottom: longitude –altitude cross section of the ozone trends at ~ 65 °N (the latitude is indicated by dashed line on top panels). Ozone trends are estimated using MEGRIDOP (left), MLS (center), and SCIAMACHY-OMPS datasets (right). For SCIAMACHY-OMPS dataset, ozone trends in the Southern Atlantic Anomaly (SAA) region are not shown, because SCIAMACHY data are flagged in this region.**

In previous studies (e.g., Petropavlovskikh et al., 2019; Steinbrecht et al., 2017; WMO, 2018), ozone trends have been evaluated at latitudes 60° S- 60° N, e.g., excluding polar regions.  In this study, we have made an attempt to evaluate ozone trends also in polar regions.

We found statistically significant positive trends in the NH polar middle stratosphere (25-30 km). In the SH polar regions, the estimated ozone trends are mostly positive, but they are not statistically significant. We found that the estimated trends at SH polar regions are sensitive to the inclusion of 2002 data into the trend analysis.  Quite exceptional (larger) ozone values in 2002 due to SH major sudden stratospheric warming, which are nearly in the beginning of our time series, result in negative, but not statistically significant, ozone trends in SH polar stratosphere, as expected.  If data from 2002 are excluded from the analysis, the trends estimates over Antarctica are not sensitive to the selection of the starting point for the trend analysis. This can be observed, for example, by comparison of ozone trends at 35 km in Figure 10 (trends in 2003-2018) and Figure 12 (trends in late 2004-2018).

Since natural variability is high in polar regions and the observation period is relatively short, it is quite expected that a simple multiple regression will lead to trend estimates that are not statistically significant. Other methods for trend analysis in polar regions, such as considering seasonal trends ( Solomon et al., 2016; Szeląg et al., 2020; Galytska et al., 2019) can be explored in future works.



## 6    Summary

In our paper, we presented the merged gridded dataset of ozone profiles (MEGRIDOP), which combines the data from
six limb-viewing satellite instruments. The merged gridded ozone profiles are the monthly means in 10°x20° latitude-longitude
bins; they cover altitudes from 10 to 50 km. The dataset covers the years 2001-2018 and will be extended regularly in the
future.

The merging was performed using aligned deseasonalized anomalies: the merged dataset represents the median of the
deseasonalized anomalies from the individual instruments. The merged deseasonalized anomalies can be used directly for
evaluation of ozone trends. For other applications, the MEGRIDOP is also available in the form of ozone number density
profiles. The dataset is available through open access at https://climate.esa.int/en/projects/ozone/data/.

The MEGRIDOP dataset can be used in different analyses. For illustration of one of the possible applications, the
climatology of ozone profiles with resolved longitudinal structure has been created. We found zonal asymmetry/structures in
the climatological ozone profiles at middle and high latitudes associated with the polar vortex. At Northern high latitudes, the
amplitude of the seasonal cycle also has a longitudinal dependence.

We evaluated regional ozone trends over the years 2001-2018 using a multiple linear regression method. Overall, the
estimated trends agree well with the trends derived from zonal mean ozone profiles.  We found a zonal asymmetry in the upper
stratospheric ozone trends at middle and high latitudes in the Northern Hemisphere: the trends are larger over
Scandinavia/Atlantic Ocean than over Siberia.  This feature agrees well with previous analyses and might be due to changes
in dynamic processed related to the Brewer-Dobson circulation.

We estimated regional and vertically resolved ozone trends also in the polar regions. As far as we know, such analysis
using limb satellite measurements is performed for the first time.  We found statistically significant positive trends in the NH
polar middle stratosphere (25-30 km). In the SH polar regions, the estimated ozone trends are mostly positive, but they are not
statistically significant.


**Acknowledgements**

The work is performed in the framework of the ESA Ozone_cci project. The GOMOS ALGOM2s dataset was created in the
framework of ESA ALGOM project. The KIT team would like to thank the European Space Agency (ESA) for giving access
to MIPAS level-1 data. The SCIAMACHY ozone retrieval was funded in parts by ESA, the German Academic Exchange
Service (DAAD) the German Aerospace Agency (DLR), and the University and State of Bremen. The data set was calculated
with resources provided by the North-German Supercomputing Alliance (HLRN). The FMI team thanks the Academy of
Finland (Project  SECTIC, and the Centre of Excellence of Inverse Modelling and Imaging). The authors thank the Canadian
Space Agency. Work at the Jet Propulsion Laboratory, California Institute of Technology, was performed under contract with
the National Aeronautics and Space Administration (NASA).



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
