# Peer review of "Measurement report: Regional trends of stratospheric ozone evaluated using the MErged GRIdded Dataset of Ozone Profiles (MEGRIDOP)"

_Atmospheric Chemistry and Physics, 2020_

## Referee Comment (RC1) · Anonymous Referee #1 · 3 Dec 2020

**1 Overall Remarks**

The paper reports on the construction of a merged long-term multi-satellite data set of ozone profiles. The data set provides monthly means on a latitude, longitude, altitude grid and covers the period 2001 to 2018. Data sets like this are very important for following and understanding the projected recovery of the ozone layer from anthropogenic ozone depleting substances, and publication in ACP is warranted. Using the merged data-set, climatological features of the global ozone distribution are presented. Decadal ozone trends are derived as a function of latitude, longitude and altitude. Data and methods of the paper are sound, and the presentation is good. I recommend

publication after addressing a few minor questions.

**2   Specific Comments**

One of the major points of the paper are longitudinal variations in both the climatological ozone distribution and in ozone trends, especially at higher northern latitudes. A good part of these variations seems related to intensity and position of the Aleutian stratospheric anti-cyclone. Unfortunately the chosen map-projection does not show this anti-cyclone very clearly. Therefore, I strongly suggest to add a few polar projection plots, especially polar projection plots that show the climatological ozone distribution along with the decadal trends for a few selected levels and months or seasons.

pg. 3 It would be good to give URLs and/or References for all the data used here, including ERA-Interim.

pg. 4, line 83: Are the used ozone profiles exactly the same as in the HARMOZ dataset, or are newer versions or reprocessed data used? Please clarify.

pg. 4, line 95: It would be good to show some plots of H. Also, for instruments with many samples (large N), the standard error off the mean might be too small / underestimated, if not all N samples are independent. The authors should probaly comment on that.

In many places, the English could be improved. The paper would benefit from copy-editing by a native speaker.

**3   Minor Comments**

pg 1, line 27: delete "areas of"?

pg. 6, line 122: replace "in two" by "between two"

pg. 6, line 123: replace "selected" by "two different longitude"

Figure 3: the high ozone in winter at 50 to 60 N, 120 to 140 E is a consequence of the Aleutian anti-cyclone. The authors might should mention that here.

pg. 7, line 142: which period? Please explain.

---

## Referee Comment (RC2) · Anonymous Referee #2 · 14 Dec 2020

The manuscript presents a new dataset (MEGRIDOP) of global stratospheric ozone vertical distribution based on the merging of various limb and occultation satellite measurements (GOMOS, SCIAMACHY, MIPAS, OSIRIS, OMPS, MLS). This dataset and its merging methodology are very similar to those SAGE–CCI–OMPS presented in Sofieva et al. (2017), except that while MLS data are included, SAGE II and ACE-FTS observations are not considered. In contrast to SAGE-CCI-OMPS, the new dataset is resolved in longitude with horizontal bins of 10°x20° in latitude and longitude. The MEGRIDOP dataset is then used for evaluating trends of stratospheric ozone as a function of longitude, latitude and altitude. The paper is well documented and provides an important contribution to the study of ozone trends in the stratosphere. It is thus suitable for publication in ACP, provided that following comments and recommendations are considered.

**Major comments**

1. The description of the merging methodology is based on the Sofieva et al. (2017) study published in ACP, itself using the results of Sofieva et al (2014) in AMT. Since the number of individual satellite datasets differs from that of SAGE-CCI-OMPS it would be interesting to see how this change in the use of satellite observations impacts the resulted latitudinal ozone fields. Such comparison could be presented in an appendix.

2. Discussion of uncertainty of the gridded monthly means from individual instruments needs improvement. It is heavily based on previous studies by the author team. Its presentation in this article is not completely self-explanatory, even if some of the equations used in the previous articles are provided here. As an example, in section 3.1, authors mention the characterization of the non-uniformity of the sampling pattern by the inhomogeneity measure H, which is a combination of asymmetry A and entropy E. But they do not precise how H is considered in the uncertainty of the averaged data and why the main contribution to H is Htime. Also, how the standard error of the mean compares with the rms of each measurement profile uncertainty? For a better understanding of uncertainty of the gridded monthly means from individual instruments, it would be useful to provide maps of H for some of the merged instruments (contrasting e.g. occultation and limb sounding instruments).

3. Evaluation of deseasonalized anomalies: for all instruments the seasonal cycle is estimated using the 2005-2011, while the 2012-2018 period is used for OMPS. This is understandable because the ENVISAT based instruments stopped in 2012. But since OMPS anomalies are adjusted to the median of anomalies from other instruments and that can impact of ozone trends, more precision of this offset as a function of altitude and latitude should be given.

4. OSIRIS and SCIAMACHY dominate the start of the record, while MLS and adjusted
OMPS dominate end of the record. How this shift in dominating instruments can impact trend as a function of altitude and latitude/longitude? As an example, Fig. 5 shows an overestimation of ozone anomalies by SCIAMACHY compared to OSIRIS and MLS. Discussion on this issue is lacking in the manuscript.

5. Section 3.3: Some discussion on possible correlation between the datasets should be provided, especially since OMPS anomalies are adjusted using the other measurements anomalies. This could affect the error bars. In equation 5, how is evaluated ïĄsïĄĎ,imed when there is an even number of measurements? Also, from the example in appendix of Sofieva et al., 2017, the final uncertainty can vary from one bin to the next, depending on the availability of data since the median is used, in particular after the stop of ENVISAT based measurements. This is illustrated in Fig. 7 that shows a decreased of uncertainty in the lower stratosphere at the end of the record in the bottom panel of the figure. How does this affect ozone trends? Even if uncertainties are not taken into account in the trend model, larger variability of data in the lower stratosphere linked to OSIRIS should affect trend results in this region. A discussion of the validity of trend results below 20 km should thus be included in the article.

6. An independent validation of the MEGRIDOP reconstructed ozone dataset (section 4) based on e.g. ground-based or other satellite instruments is lacking. Validation using ozone sondes (up to 25 – 30km) as well as SAGE II (up to 2004) or more recent ISS/SAGE III data would be an asset for the study.

7. The latest compilation of stratospheric ozone trends from Petropavlovkikh et al. (2019) emphasises the lack of significant ozone trends in the lower stratosphere, pointing to a potential discrepancy with results from CCI models, although not significant at 2 sigma level. Other publications have also addressed ozone trends in the lower stratosphere (Ball et al., 2018; 2019; Wargan et al., 2018). Considering the importance of this issue, a dedicated paragraph addressing ozone trends in the lower stratosphere should be added. Such a discussion could include quantification of ozone trends in the lower stratosphere in the SH high latitudes, in order to eventually confirm ozone
recovery in this region (e.g. Salomon et al., 2016).

8. Figure 10 and 12 show different trend results from the MEGRIDOP dataset at 35 km, with more pronounced positive ozone trends in the tropics in the period 2004 - 2018 compared to the period 2003 - 2018. Such a sensitivity to the starting year is interesting. Can the authors comment on that? Also on the non-significant decrease of ozone over Siberia at 20, 25 and 35 km. The asymmetry of trends between the Northern and Southern hemispheres at 20 and 25 km deserves also some discussion.

**References**

Ball, W. T., et al., Evidence for a continuous decline in lower stratospheric ozone offsetting ozone layer recovery, Atmos. Chem. Phys., 18(2), 1379–1394, doi:10.5194/acp-18-1379-2018, 2018.

Ball, W. T., Alsing, J., Staehelin, J., Davis, S. M., Froidevaux, L., and Peter, T.: Stratospheric ozone trends for 1985–2018: sensitivity to recent large variability, Atmos. Chem. Phys., 19, 12731–12748, https://doi.org/10.5194/acp-19-12731-2019, 2019.

Solomon, S., Ivy, D. J., Kinnison, D., Mills, M. J., Neely, R. R. and Schmidt, A.: Emergence of healing in the Antarctic ozone layer, Science (80)., 353(6296), 269–274, doi:10.1126/science.aae0061, 2016.

Wargan, K., et al.: Recent decline in extratropical lower stratospheric ozone attributed to circulation changes, Geophys. Res. Lett., 45, 5166–5176, https://doi.org/10.1029/2018GL077406, 2018.

Specific comments

L75: It is not clear why the authors use MLS temperatures for conversion to ozone number density but ERA-Interim data for altitude-pressure conversion. Did the author check sensitivity of the results using ERA-Interim data for number density? ERA-Interim data stop in August 2019. They are now replaced by ERA-5. Is there a prospect to use ERA-5 for extending the MEGRIDOP dataset to 2020 and beyond? **ACPD**
L103: Fig 1 as well as Fig. 2 and all similar color figures lack axis titles.

L137: The use of deseasonalized anomalies enables the removing of biases if sampling patterns do not change over time. Is it true? Can the authors comment on this?

L148: Fig. 5 lacks the median curve.

L157: In equation 3, the term ïĄši is missing (using error propagation). The term "relative" should be added to uncertainty.

---

## Author Comment (AC2) · 1 Mar 2021

Dear Reviewer,

Thank you very much for your comments on our manuscript. We took your comments into account in the revised version of the manuscript. Please find below our detailed replies (black font) on your comments (blue font).

Reviewer #2

Major comments
1. The description of the merging methodology is based on the Sofieva et al. (2017) study published in ACP, itself using the results of Sofieva et al (2014) in AMT. Since the number of individual satellite datasets differs from that of SAGE-CCI-OMPS it would be interesting to see how this change in the use of satellite observations impacts the resulted latitudinal ozone fields. Such comparison could be presented in an appendix.

Authors:
The sensitivity of the merging method to the number of instruments (including the influence on trends, with several illustrations) is studied in details in (Sofieva et al., 2017) and its Supplement. We found only minor changes in ozone trends after 1997 caused by variations in number of instruments. We would like to note that the longitudinally resolved MEGRIDOP, in addition to inclusion of MLS data, covers different time period compared to the zonally averaged SAGE-CCI-OMPS dataset, and thus the trend analyses are different.
In the revised version, after Eq.(2), we added: " The advantage of using the median estimate is that the merged anomaly follows the majority of the data, and it is not very sensitive to exclusion/addition of an individual data record, in cases where there are several (and consistent) anomaly datasets available. The sensitivity of the dataset and the evaluated trends to the number of instruments was studied in detail for SAGE-CCI-OMPS dataset, which is created with the same merging algorithm (see Sofieva et al., 2017 and its Supplements), and this is valid also for MEGRIDOP."

Reviewer #2
2. Discussion of uncertainty of the gridded monthly means from individual instruments needs improvement. It is heavily based on previous studies by the author team. Its presentation in this article is not completely self-explanatory, even if some of the equations used in the previous articles are provided here. As an example, in section 3.1, authors mention the characterization of the non-uniformity of the sampling pattern by the inhomogeneity measure H, which is a combination of asymmetry A and entropy E. But they do not precise how H is considered in the uncertainty of the averaged data and why the main contribution to H is Htime. Also, how the standard error of the mean compares with the rms of each measurement profile uncertainty? For a better understanding of uncertainty of the gridded monthly means from individual instruments, it would be useful to provide maps of H for some of the merged instruments (contrasting e.g. occultation and limb sounding instruments).

Authors:
An illustration of inhomogeneity measures $H$ is included in the Supplement. In the text of the revised version we also added a note about typical values of $H$. We indicate in the paper (also in the original version) that the main contribution is $H_{time}$. We use $H_{time}$ for detection of spatial bins with high levels of data inhomogeneity. In the revised version, it is indicated not only in Sect. 3.2, but also in Sect. 3.1.

For our application, we do not see what additional information is obtained by comparing the standard error of the mean to uncertainties in individual ozone profiles – they characterize different parameters.

Reviewer #2
3. Evaluation of deseasonalized anomalies: for all instruments the seasonal cycle is estimated using the 2005-2011, while the 2012-2018 period is used for OMPS. This is understandable because the ENVISAT based instruments stopped in 2012. But since OMPS anomalies are adjusted to the median of anomalies from other instruments and that can impact of ozone trends, more precision of this offset as a function of altitude and latitude should be given.

Authors:

We would like to note that adjustment of deseasonalized anomalies is performed, not anomalies (The original text is "First, we offset the OMPS deseasonalized anomalies to the median of the deseasonalized anomalies from all other instruments"). The alignment of deseasonalized anomalies is a general procedure (like bias correction). The offset is evaluated using a sufficiently long time period (6 years), so "an impact of ozone trends" is not expected. These offsets are pure technical information, which characterizes neither the quality of OMPS data (the offsets are mainly related to the difference in seasonal cycles 2005-2011 and 2012-2018) nor the uncertainty/stability of the merged dataset. Therefore, we believe that a detailed illustration of its three-dimensional structure is not needed.

Reviewer #2
4. OSIRIS and SCIAMACHY dominate the start of the record, while MLS and adjusted OMPS dominate end of the record. How this shift in dominating instruments can impact trend as a function of altitude and latitude/longitude? As an example, Fig. 5 shows an overestimation of ozone anomalies by SCIAMACHY compared to OSIRIS and MLS. Discussion on this issue is lacking in the manuscript.

Authors:
We cannot agree with the above formulation. First, OSIRIS and MLS data are present during the whole /nearly whole time period. Second, since the merged anomaly is the median of individual anomalies and since the individual anomalies are very close to each other, it is impossible to clearly identify the "dominating instrument(s)". For example, in the study related to the SAGE-CCI-OMPS, which uses the same merging principle, the representativeness of individual datasets in the merged dataset is studied (see Supplement to Sofieva et al., 2017). It is shown there that the deseasonalized anomalies from individual datasets are usually very close to each other, so that several values can be typically found within the uncertainty interval of the merged anomaly $\Delta_{merged} \pm \sigma_{\Delta, merged}$. This is true also for MEGRIDOP.
In the revised version of the manuscript, we highlighted this.

Reviewer #2
5. Section 3.3: Some discussion on possible correlation between the datasets should be provided, especially since OMPS anomalies are adjusted using the other measurements anomalies. This could affect the error bars.

Authors:
Off-setting does not increase correlation.   However, deseasonalized anomalies from individual instruments are highly correlated: they describe the same natural ozone variations.

That is why we characterize the uncertainty by Eq.(5)

In equation 5, how is evaluated ï A̧sï A̧D,imed when there is an even number of measurements?

In case of even number of measurements, the mean of two neighbors to the median is used. This is the standard procedure. We clarified this in the revised version.

Also, from the example in appendix of Sofieva et al., 2017, the final uncertainty can vary from one bin to the next, depending on the availability of data since the median is used, in particular after the stop of ENVISAT based measurements. This is illustrated in Fig. 7 that shows a decreased of uncertainty in the lower stratosphere at the end of the record in the bottom panel of the figure. How does this affect ozone trends? Even if uncertainties are not taken into account in the trend model, larger variability of data in the lower stratosphere linked to OSIRIS should affect trend results in this region. A discussion of the validity of trend results below 20 km should thus be included in the article.

We confirm that the uncertainties are not used as weights in the regression model. In the revised version, we added: "The uncertainties for the merged data are not used in the regression analysis as weights: different amounts of data available over time result in varying uncertainties over time (e.g., as shown in Figure 7), which might improperly weight the time series. In our regression, all data points are considered with equal weights, and the uncertainty of the fitted parameters is estimated from the regression residuals."

Larger uncertainties do not necessary imply larger data variability. Related to your question/example of the UTLS, the typical UTLS values of estimated uncertainties are in the range of 2-12 % before 2012 and 2-6 % after 2012, which is significantly smaller than the natural variability in the UTLS, which is typically tens of percent (up to 100 % in the tropical UTLS).
A discussion of trend results below 20 km is nevertheless important, and we added it in the revised version.
We added also typical values of estimated uncertainties in the stratosphere and in the UTLS in Sect. 3.3

Reviewer #2
6. An independent validation of the MEGRIDOP reconstructed ozone dataset (section 4) based on e.g. ground-based or other satellite instruments is lacking. Validation using ozone sondes (up to 25 – 30km) as well as SAGE II (up to 2004) or more recent ISS/SAGE III data would be an asset for the study.

Authors:
Validation usually means comparison with the reference dataset, which has a known high quality. We would like to note that there are numerous studies of comparison of collocated ozone profiles from individual satellite instruments with ground-based and other satellite data. We would like to emphasize that MEGRIDOP represent the monthly zonal mean ozone profiles in 10°×20° latitude - longitude bins. There is no obvious way to validate the derived dataset with reference to sondes or SAGE in a meaningful manner (i.e., one from which quantitative conclusions as to "validity" can usefully be drawn), given the inherent vast disparity in spatial and temporal coverage and thus representativeness.
We think that MEGRIDOP can be used for validation/intercomparisons of climate data records from ground-based and satellite measurements. In the revised version, we added this as a suggestion of future analyses using MEGRIDOP.

Reviewer #2
7. The latest compilation of stratospheric ozone trends from Petropavlovkikh et al. (2019) emphasises the lack of significant ozone trends in the lower stratosphere, pointing to a potential discrepancy with results from CCMI models, although not significant at 2 sigma level. Other publications have also addressed ozone trends in the lower stratosphere (Ball et al., 2018; 2019; Wargan et al., 2018). Considering the importance of this issue, a dedicated paragraph addressing ozone trends in the lower stratosphere should be added. Such a discussion could include quantification of ozone trends in the lower stratosphere in the SH high latitudes, in order to eventually confirm ozone recovery in this region (e.g. Salomon et al., 2016).

Authors:
The trends in the lower stratosphere, including the trends in polar regions, are discussed now in our paper in more detail.

Reviewer #2
8. Figure 10 and 12 show different trend results from the MEGRIDOP dataset at 35 km, with more pronounced positive ozone trends in the tropics in the period 2004 – 2018 compared to the period 2003 – 2018. Such a sensitivity to the starting year is interesting. Can the authors comment on that?

Authors:
A remarkable sensitivity of tropical ozone trends at ~35 km to the selection of the period for evaluation of ozone trends has been reported in several papers (e.g., Arosio et al., 2019; Galytska et al., 2019; Laine et al., 2014). This might be related to a decadal-scale $O_3$ oscillation resulting from changes in Brewer-Dobson Circulation.
In the revised version, we added a corresponding note.

Also on the non-significant decrease of ozone over Siberia at 20, 25 and 35 km. The asymmetry of trends between the Northern and Southern hemispheres at 20 and 25 km deserves also some discussion.

In the revised version, we added the note about the difference of trend in the Northern and Southern hemispheres.

Reviewer #2
Specific comments
L75: It is not clear why the authors use MLS temperatures for conversion to ozone number density but ERA-Interim data for altitude-pressure conversion. Did the author check sensitivity of the results using ERA-Interim data for number density?

Since all reanalyses data may have artificial jumps due to different amount of assimilated data (e.g., (Simmons et al., 2014) and thus they are, in general, not designed for trend analysis, we think that it is preferable to use observations, when possible. In addition, MLS observes nearly the same air masses at the same time for its temperature and ozone measurements.

ERA-Interim data stop in August 2019. They are now replaced by ERA-5. Is there a prospect to use ERA-5 for extending the MEGRIDOP dataset to 2020 and beyond?

Yes, we are planning to extend the MEGRIDOP; in this extension, ERA-5 will be used. This is mentioned in the revised version of the paper.

L103: Fig 1 as well as Fig. 2 and all similar color figures lack axis titles.

Since coast lines are added in the figures, additional axis titles are not needed.

L137: The use of deseasonalized anomalies enables the removing of biases if sampling patterns do not change over time. Is it true? Can the authors comment on this?

We made this statement more accurate in the revised version: "The main advantage of using deseasonalized anomalies is that various biases between the individual datasets - e.g., instrumental-specific, or those due to the difference in local time - are automatically removed. The deseasonalization also removes spatial sampling biases if the sampling patterns do not change over time. "

L148: Fig. 5 lacks the median curve.

This is done intentionally, in order to visualize clearly the OMPS off-setting.

L157: In equation 3, the term ïA˛ši is missing (using error propagation). The term "relative" should be added to uncertainty.

Thank you, the misprint is corrected.

**References:**
Arosio, C., Rozanov, A., Malinina, E., Weber, M. and Burrows, J. P.: Merging of ozone profiles from SCIAMACHY, OMPS and SAGE II observations to study stratospheric ozone changes, Atmos. Meas. Tech., 12(4), 2423–2444, doi:10.5194/amt-12-2423-2019, 2019.

Galytska, E., Rozanov, A., Chipperfield, M., Dhomse, S., Weber, M., Arosio, C., Wuhu, F. and Burrows, J.: Dynamically controlled ozone decline in the tropical mid-stratosphere observed by SCIAMACHY, Atmos. Chem. Phys., 19, 767–783, doi:10.5194/acp-19-767-2019, 2019.

Laine, M., Latva-Pukkila, N. and Kyrölä, E.: Analysing time-varying trends in stratospheric ozone time series using the state space approach, Atmos. Chem. Phys., 14(18), 9707–9725, doi:10.5194/acp-14-9707-2014, 2014.

Simmons, A. J., Poli, P., Dee, D. P., Berrisford, P., Hersbach, H., Kobayashi, S. and Peubey, C.: Estimating low-frequency variability and trends in atmospheric temperature using ERA-Interim, Q. J. R. Meteorol. Soc., 140(679), 329–353, doi:10.1002/qj.2317, 2014.

Sofieva, V. F., Kyrölä, E., Laine, M., Tamminen, J., Degenstein, D., Bourassa, A., Roth, C., Zawada, D., Weber, M., Rozanov, A., Rahpoe, N., Stiller, G., Laeng, A., von Clarmann, T., Walker, K. A., Sheese, P., Hubert, D., van Roozendael, M., Zehner, C., Damadeo, R., Zawodny, J., Kramarova, N. and Bhartia, P. K.: Merged SAGE II, Ozone_cci and OMPS ozone profile dataset and evaluation of ozone trends in the stratosphere, Atmos. Chem. Phys., 17(20), 12533–12552, doi:10.5194/acp-17-12533-2017, 2017.

---

## Author Comment (AC1)

Dear Reviewer,

Thank you very much for your positive evaluation and comments on our manuscript. We took your comments into account in the revised version of the manuscript. Please find below our detailed replies (black font) on your comments (blue font).

Reviewer #1

**2 Specific Comments**
One of the major points of the paper are longitudinal variations in both the climatological ozone distribution and in ozone trends, especially at higher northern latitudes. A good part of these variations seems related to intensity and position of the Aleutian stratospheric anti-cyclone. Unfortunately, the chosen map-projection does not show this anti-cyclone very clearly. Therefore, I strongly suggest to add a few polar projection plots, especially polar projection plots that show the climatological ozone distribution along with the decadal trends for a few selected levels and months or seasons.

Authors:
We plotted the climatology distributions and the trends also in polar projections (for both hemispheres) and included these figures in the Supplement. We do not observe a clear relation of ozone distribution and trends to the position of the Aleutian anti-cyclone. The observed regional trends might be related to the average position of the polar vortex. However, more detailed analyses are needed in order to confirm/reject this hypothesis. Such future analyses might include, for example, analyses of seasonal dependence of ozone trends or winter-spring trends related to the position of the polar vortex. This is discussed in slightly more detail in the revised version of our paper.

Reviewer #1
pg. 3 It would be good to give URLs and/or References for all the data used here, including ERA-Interim.

Authors: In Table 1, the references to the publications describing the individual datasets are collected. In the revised version, we added the reference to the ERA-Interim data and updated the reference to the HARMOZ_ALT dataset.

Reviewer #1
pg. 4, line 83: Are the used ozone profiles exactly the same as in the HARMOZ dataset, or are newer versions or reprocessed data used? Please clarify.

Authors: it is the updated HARMOZ dataset, we indicated this in the revised version.

Reviewer #1
pg. 4, line 95: It would be good to show some plots of H. Also, for instruments with many samples (large N), the standard error off the mean might be too small / underestimated, if not all N samples are independent. The authors should probably comment on that.

Authors:
An illustration of inhomogeneity measures $H$ is now included in the Supplement. In the revised version, we also added a note: "The spatial bins are covered rather uniformly by the data. The inhomogeneity measure $H$ is very close to zero for the instruments with dense sampling (MIPAS, SCIAMACHY, MLS,

OMPS). For OSIRIS and GOMOS, *H* is usually below 0.1 (good homogeneity of the data) with a few exceptions for some months and locations"

We added also a caveat about possible influence of correlations caused by orbital sampling on the standard error of the mean estimate with the reference to Toohey and von Clarmann (2013), and indicated that this is an approximation.

Reviewer #1
In many places, the English could be improved. The paper would benefit from copyediting by a native speaker.

Authors: The paper has been improved and will also be corrected by professionals who have a formal education in the English language.

Reviewer #1
**3 Minor Comments**
pg 1, line 27: delete "areas of"?

pg. 6, line 122: replace "in two" by "between two"
pg. 6, line 123: replace "selected" by "two different longitude"

Corrected

Figure 3: the high ozone in winter at 50 to 60 N, 120 to 140 E is a consequence of the Aleutian anti-cyclone. The authors might should mention that here.

As written above, we do not observe a clear relation with the Aleutian anticyclone.

pg. 7, line 142: which period? Please explain.

Corrected